# Assessment of Energy Expenditure of Police Officers Trained in Polish Police Schools and Police Training Centers

**DOI:** 10.3390/ijerph19116828

**Published:** 2022-06-02

**Authors:** Jerzy Bertrandt, Anna Anyżewska, Roman Łakomy, Tomasz Lepionka, Ewa Szarska, Andrzej Tomczak, Agata Gaździńska, Karolina Bertrandt-Tomaszewska, Krzysztof Kłos, Ewelina Maculewicz

**Affiliations:** 1Faculty of Economic Sciences, John Paul II University of Applied Sciences in Biała Podlaska, Sidorska 95/97, 21-500 Biała Podlaska, Poland; 2University of Economics and Human Sciences in Warsaw, Okopowa 59, 01-043 Warsaw, Poland; a.anyzewska@vizja.pl; 3Department of Hygiene and Physiology, Military Institute of Hygiene and Epidemiology, 4 Kozielska, 01-163 Warsaw, Poland; roman.lakomy@interia.pl (R.Ł.); tomasz.lepionka@wihe.pl (T.L.); eszarska@gmail.com (E.S.); karolinabertrandt@gmail.com (K.B.-T.); 4Independent Researcher, 02-348 Warsaw, Poland; biuro.at@onet.pl; 5Department of Psychophysiological Measurements and Human Factor Research, Military Institute of Aviation Medicine, 54/56 Krasinskiego, 01-755 Warsaw, Poland; afrotena@gmail.com; 6Department of Infectious Diseases and Allergology, Military Institute of Medicine, Szaserów 128, 04-141 Warsaw, Poland; kklos@wim.mil.pl; 7Faculty of Physical Education, Jozef Pilsudski University of Physical Education in Warsaw, 34 Marymoncka, 00-809 Warsaw, Poland; ewelina.maculewicz@awf.edu.pl

**Keywords:** energy expenditure, police schools, police officers, physical activity, hardness of work

## Abstract

Knowledge of the energy expenditure related to the training of policemen allows for assessment of the intensity of the work performed and is an indispensable element of planning and implementing nutrition. This study on energy expenditure comprised a total of 280 persons, students of two Polish police schools and two police training centers. The energy expenditure of policemen was determined based on measurements of the heart rate using Polar RC3 GPS heart rate monitors. The energy expenditure of policemen associated with the training process in the police schools and training centers ranged from 1793 to 3043 kcal/8 h and amounted to 2314 ± 945 kcal/8 h during training on average. The values of energy expenditure related to a typical training day in Polish institutions conducting police training are diverse and depend on the specificity and nature of the training. According to the criteria for assessing the burden of work, the work performed by police officers can be classified as hard work and very hard work.

## 1. Introduction

Many studies have concluded that physical inactivity is a primary cause of most chronic diseases [1,2,3]. The benefits of physical activity for maintaining health have been well documented, especially in the prevention and treatment of chronic diseases such as certain cancers, type 2 diabetes, and cardiovascular diseases [4,5]. This is especially important in the police force, as law enforcement officers are often required to adapt quickly from sedentary, passive functions to hostile environments where maximum body effort is needed [6]. In this context, accurate measurement of energy expenditure is essential for both epidemiological studies and assessment of human nutritional needs [7,8].

Total energy expenditure is the energy required by the body during a 24-h period and is determined by the sum of three components: basal energy expenditure, diet-induced thermogenesis, and physical activity [9]. Determination of energy expenditure is important to adjust the nutritional habits of individual people and must take into account the energy requirements for physical activity and specific health conditions. To date, no research has been conducted in Poland on the energy burden associated with the service and training of officers studying at police schools and trained at police training centers. In the Polish uniformed services, such research was conducted only among students of various types of military schools and fire schools. These studies indicate that students at military universities were burdened with an energy expenditure of 3339 to 4121 kcal/d, while students at fire schools—from 3735 to 4745 kcal/d [10,11,12,13]. According to the workload classification by Lehman, they performed medium to very heavy work [14].

The issues related to energy expenditure, the energy value of food, and, consequently, the systemic energy balance are of particular importance in the uniformed services. Knowledge of the physical burden connected with the specificity of the service performed should be an indispensable element of food planning, which must cover the energy needs of the body and provide all necessary nutrients in the right amounts and proportions. The police are a uniformed and armed formation serving the public and intended to protect human security and to maintain public safety and order. Knowledge of the energy expenditure related to the training of police officers allows for assessment of the severity of work performed and provides a possibility to quantify it and forms the basis for establishing nutrition standards. The heaviness of work, the measure of which is the value of energy expenditure, is an indispensable element of an assessment of physical load in accordance with psychophysical capabilities, while the physical load is a relation between the requirements of work based on physical effort and the capabilities of the body.

The aim of this study was to determine the energy expenditure of students at police schools and police training centers in relation to the specificity and nature of the training to assess degree of intensity of their work.

## 2. Materials and Methods

### 2.1. Participants

The research on energy expenditure covered 270 students at four police training institutions: two police schools, a police training center, and a police prevention department.

The examination of energy expenditure covered 280 persons trained in 2 Polish police schools and 2 police training centers. The research involved 60 students at the police academy, including 50 men and 10 women; 113 students, all men, trained on specialist courses at the police training center; 47 students at the police school; and 60 policemen trained at the police prevention department.

Due to the small number of women studying in police schools and police training centers, only men took part in the research. Thus, the study covered 50 students of the police academy, 113 students trained in specialist courses at the police training center, 47 students at the police school, and 60 students at the police prevention department. Among the 113 students of the police training center, 74 people were trained on a specialist course for police officers intervening against aggressive and dangerous persons, 18 participated in training for future police dog handlers, and 21 were trained to work in the water police. All students participated in the same theoretical classes and classes on the use of weapons, while the other specialist classes took into account the specificity of the future service in the police.

The research on the students of the police academy and the police school was conducted during typical training activities included in the training plan and covered both theoretical classes in the form of lectures and those related to high physical activity, such as tactics or shooting.

The research was conducted in accordance with the Helsinki Declaration of the World Medical Association and was approved by the Ethics Committee of the Military Institute of Hygiene and Epidemiology (no. 1/XXI 95/2016). Participants received an information sheet about the details of the study, the purpose, and the procedures used as well as potential risks and benefits of their participation.

### 2.2. Measurement of Height and Weight

Body height (without shoes) was measured using a portable stadiometer (TANITA HR-001, Tanita Corporation, Tokyo, Japan). Body weight was measured using a bioelectrical impedance analysis (BIA) with the TANITA MC-780 103 device (Tanita Corporation, Tokyo, Japan), with an accuracy of 0.1 kg, according to the procedure specified in the instruction manual (lightly dressed, without shoes). All measurements were performed according to the procedure specified in the instruction manual and without any metal objects.

### 2.3. Measurement of Energy Expenditure

The method of analyzing heart rate changes was used to measure energy expenditure. The energy expenditure study included heart rate measurements with a Polar RC3 GPS heart rate monitor (Polar Electro Oy, Kempele, Finland). The values of energy expenditure of the activities performed by the policemen constitute the average value of at least three measurements. The obtained results were the basis for determining the energy demand of policemen in relation to the specificity and nature of training and/or service, as well as for the assessment of the intensity of work, in accordance with the classification given by Lehman (Table 1) [14].

### 2.4. Statistical Analyses

All statistical analyses were performed using the program R (The R Foundation for Statistical Computing v2.0–1. https://cran.r-project.org, accessed on 20 September 2021). Anthropometric data are shown as mean values ± standard deviation, and differences among experimental groups were analyzed with Student’s *t*-test, which were statistically significant when *p* < 0.05. To check the compliance of the variables with the normal distribution, the Shapiro–Wilk test was used, and Levene’s test was used for verification of the homogeneity of variance. The differences in the values of energy expenditure of students at individual universities and training centers were calculated using the chi-square test. The data spread is presented in interquartile range (IQR) values.

## 3. Results

### 3.1. Characteristics of the Studied Groups

The characteristics of the studied groups are given in Table 2.

In the police academy, the research included students of the police service preparation course.

The group of people trained in the police prevention department was the youngest, which resulted from the student selection criteria, while students at the police school were characterized by the highest body weight.

### 3.2. Energy Expenditure of Police Academy Students

The study of students’ energy expenditure was carried out during the implementation of tasks in the study program at the police academy. It included measurements both during typical theoretical classes and in field conditions, during which students learned how to use weapons, arrest procedures, chase, etc. The results of the energy expenditure of police officers carrying out typical training tasks during 8 h of training are presented in Table 3.

Analysis of the obtained results showed that the average energy load of police officers carrying out training tasks at the basic course in the police academy amounted to 2233 ± 546 kcal/8 h and varied depending on the activities performed. This value, according to Lehman’s classification of workload, allows to classify the work performed as very heavy work [6].

### 3.3. Energy Expenditure of Students at the Police Training Center

#### 3.3.1. Energy Expenditure of Students Trained on Specialist Courses for Police Officers

The research on energy expenditure covered 74 men, all students at the police training center trained on specialist courses for police officers intervening against aggressive and dangerous people and for instructors of police shooting in anti-terrorist police units. The mean body weight and mean height of the officers were 90.4 ± 13.8 kg and 179.7 ± 7.8 cm, respectively. The energy expenditure of police officers trained for the above-mentioned specialist courses included both training tasks of a theoretical nature (lectures) and those related to high physical activity, e.g., preparation and implementation of shooting. The results of the energy load of officers in relation to the training process are summarized in Table 4.

It was shown that the average energy expenditure of an officer associated with a typical 8-h day of program training was 2458 ± 723 kcal, which places the work performed in the category of very heavy work. It should be emphasized that there is a large variation in the results of energy load related to the implementation of various types of training. Theoretical training resulted in a low energy load of students, amounting to 1.75 kcal/min, which was characteristic of light work, while during intensive tactical classes, it amounted to 7.67 kcal/min, indicating the performance of hard work.

#### 3.3.2. Energy Expenditure of Officers Trained as Future Service Dog Handlers

Another assessment concerned the energy expenditure of officers trained as future service dog handlers. The study involved 18 officers, aged 34 ± 4.3 years, of an average body weight of 83.2 ± 16.0 kg and body height of 174.0 ± 7.0 cm. The training included training of patrol dogs and dogs for special tasks such as searching for drugs or explosives. The results of the energy expenditure of service dog handlers are summarized in Table 5.

The energy expenditure related to the training of service dog handlers amounted to 2111 ± 834 kcal/8 h, which puts the severity of work performed in the category of very heavy work.

#### 3.3.3. Energy Expenditure of Police Officers Who Perform Tasks on Waters

The police training center trains, among others, police officers who perform tasks on waters and in surrounding areas. During these courses, students acquire skills related to performing specialized tasks in the preventive service on water; rescuing and searching for people, property, and floating equipment; and maneuvering a boat in difficult and extreme weather conditions. They also acquire skills in organizing rescue operations, handling specialized rescue and navigation equipment, and learning to use modern means of transport, including water scooters, while on patrol. The training process takes place mainly on water and concerns driving and operating motorboats and navigation and rescue operations in water areas.

The study of energy expenditure of police officers connected with the specificity of this training covered 21 male officers, aged 34.5 ± 6.6 years, with an average body weight of 89.5 ± 10.9 kg and height of 184.5 ± 4.5 cm. The values of energy load associated with the training are summarized in Table 6.

It was shown that an 8-h day of training resulted in an energy load of 1973 ± 553 kcal, indicating that they were performing heavy work. The relatively low energy expenditure of the water police trainees compared to those participating in other courses results from the specificity of patrolling water areas on motorboats, which is related to their low physical activity. Analysis of the average values of energy expenditure incurred by officers during the 8-h training process in the three centers included in the police training center showed that officers were burdened with different levels of energy expenditure, which resulted from the specificity of the training. The highest value of energy load related to the 8-h training day was found in officers trained on specialist courses and during the training of service dog handlers, which qualified the work as very hard work, while officers trained on water performed heavy work.

### 3.4. Energy Expenditure of Policemen Trained in the Police School

The police school specializes in training prevention police officers, i.e., those whose service has a direct impact on public order and the sense of security of citizens. Among other things, the school provides basic vocational training that every police officer admitted to the service must undergo. It prepares them for the implementation of tasks in basic executive positions (e.g., in patrol and intervention services, in police prevention units, or in a convoy service). Students gain knowledge and skills in the fields of law, crime prevention, forensics, intervention tactics and techniques, psychology, social communication, ethics, human rights, first aid, shooting training, and operation of IT and communication equipment. They learn how to perform patrol intervention and convoy protection services as well as the specifics of work as a district policeman and unit officer on duty.

A total of 47 male officers with an average body weight and height of 93.8 ± 15.3 kg and 180.4 ± 6.9 cm, respectively, were included in this energy expenditure study.

The results of the research on the energy load of officers during the 8-h training day are summarized in Table 7.

The results of the research on the energy expenditure of officers trained in the police school show that the work performed during the training is very hard work.

### 3.5. Energy Expenditure of Policemen Trained in the Police Prevention Unit

Police prevention units are designed mainly for team activities within compact subunits. The main tasks of these units include the following:-Protection of public security and order during legal gatherings and mass events;-Restoring public order in the event of collective violation of the law;-Protection of public order in the event of constitutionally defined states of emergency as well as catastrophes and natural disasters;-Pursuit of dangerous criminals.

The study involved 60 officers with an average body weight of 83.7 ± 12.5 kg and an average body height of 180.3 ± 6.5 cm. The research included activities carried out on a typical training day, including combat tactics, drills, and shooting.

The obtained results are summarized in Table 8.

It should be noted that in all examined police units, there was a large variation in the results, from low values of 1463.0 ± 339.0 kcal/8 h in the case of women’s training up to high values of 3043 kcal/8 h, which was connected with the specificity, conditions, and nature of the training and service performed. The average values of energy expenditure of the police officers trained for 8 h in all divisions are summarized in Table 9.

The analysis of the obtained values of energy load of the examined students in relation to the training processes showed that their energy expenditure from the implementation of individual training tasks was similar and ranged from 0.42 to 0.71 kcal/min/kg bw (Table 10). Although the largest energy expenditure related to the implementation of individual training activities, as well as that related to the 8-h training process, was observed in the students of police prevention units, it was not a statistically significant difference. On the other hand, the lowest energy expenditure related to the training was observed in the water police.

## 4. Discussion

Poland is one of the few countries where determination of the energy expenditure of workers on tasks at work is required by law. Information on how physically demanding work is (a measure of energy expenditure) at each workstation is essential not only for comparison with applicable regulations concerning maximum allowable values for regular work activity but also for work planning and taking proactive action to reduce the adverse health effects of work.

The value of energy expenditure related to an 8-h work shift should be taken into account when planning work, planning breaks, and allocating preventive meals and drinks by the employer. Heavy workload increases the risk of musculoskeletal system dysfunction, which is one of the causes of accelerated degenerative changes (especially of the spine) and an accelerated decline in exercise capacity. Hard physical work should also be considered a risk factor for cardiovascular disorders such as high blood pressure and ischemic heart disease. The assessment of energy expenditure, and thus the severity of work performed, allows to quantify the physical load of an employee according to his/her endurance capabilities.

Law enforcement is a highly stressful occupation that is prone to increasing the prevalence and incidence of cardiovascular disease. Evidence indicates that the prevalence of traditional cardiovascular risk factors among police officers is high (often higher than in the general population). Police work creates exposure to risk factors for the development of cardiovascular disease and diabetes and results in increased mortality rates [15,16,17]. Epidemiological studies suggest that police officers and related public security personnel develop an increased risk of cardiovascular morbidity and mortality. Currently employed police personnel have a high prevalence of traditional risk factors, including hypertension, hyperlipidemia, metabolic syndrome, smoking, and a sedentary lifestyle. Moreover, low physical activity of policemen leads to a positive energy balance and, consequently, to obesity [18,19]. Obesity may be more common among police officers compared with civilians, whereas diabetes is present less frequently. Law enforcement personnel are also exposed to occupational risk factors such as sudden physical exertion, acute and chronic psychological stress, shift work, and noise [16]. Obesity not only affects the ability of police officers to perform their work-related duties, but consequently, it may also impact public safety.

In the available literature, there are few works concerning the energy expenditure of students at police schools and police officers trained in police training centers. The values of energy expenditure obtained in the present study indicate that the energy loads of students and trainees participating in training programs in Polish police schools and police training centers range from 1973 ± 553 to 3043 ± 1308 kcal/8 h of work. These values place their work in the category of very hard work.

Studies in police schools can be compared to studies from fire service schools. The results of earlier research revealed that the values of energy expenditure during typical activities from the training program prepared for students at the Main School of Fire Service were diverse and ranged from 1.49 to 10.66 kcal/min. According to Christensen’s classification of work severity, the work performed by students can be classified as light work to very heavy work [20]. The average daily energy load of students at the Main School of Fire Service on a typical day of training on the training ground was 4745 ± 1181 kcal/d, which means that the work performed should be considered, according to the obligatory classification of work intensity, as very heavy [11].

Previous studies on the energy load of students at the National Fire Service Aspirants School showed that their daily energy expenditure from the training process amounted to 3735.5 kcal, while during 8 h of program classes, students expended only 1289.5 kcal [12].

Historical studies on the energy burden of 30 male and 10 female police cadets, randomly selected from the Singapore Police Force, showed that the value of daily energy expenditure associated with duty and training tasks was 3028 and 1752 kcal, respectively [21].

A study on the energy expenditure of 28 healthy police officers working in shifts showed that the energy expenditure was 3062 kcal/d during night shifts, amounted to 2647 kcal/d during day shifts, and was the lowest on holiday duty, when it amounted to 2310 kcal/d [22].

An assessment of the service-related daily energy expenditure of Malaysian police officers showed that males expended 2639.6 ± 229.4 kcal/d and females 2268.9 ± 203.5 kcal/d [23].

The values obtained in the present study on energy expenditure associated with studies and training in Polish police schools and police training centers confirm the results of previous research, indicating that both studies and police training as well as police service are characterized by work that falls into the category of heavy work.

## 5. Conclusions

Assuming that the average value of men’s energy expenditure during the training process in Polish police schools and the police training center, as well as in the prevention unit, amounts to 2314 ± 945 kcal/8 h of training, it should be concluded that the work performed by police officers belongs to the category of very hard work.The energy expenditure related to the implementation of the study and training program depends on the type of university or training center as well as the specificity of the performed training activities.The energy value of the daily food ration used in the nutrition of police officers trained in Polish police schools and police training centers should be adjusted to their energy expenditure.

## Figures and Tables

**Table 1 ijerph-19-06828-t001:** Classification of work severity on the basis of energy expenditure values for an 8-h working day [6].

Severity of Work	Energy Expenditure during an 8-h Working Day			
	Male		Female	
	kcal	kJ	kcal	kJ
Very light	<300	<1256	<200	<837
Light	300–800	1256–3350	200–700	837–2930
Moderate	800–1500	3350–6280	700–1000	2930–4187
Hard	1500–2000	6280–8374	1000–1200	4187–5024
Very hard	>2000	>8374	>1200	>5024

**Table 2 ijerph-19-06828-t002:** Characteristics of police officers.

	Students of the Police Academy	Students of the Police Training Center	Students of the Police School	Policemen Trained in the Police Prevention Department
Sex	Male—50	Male—113	Male—47	Male—60
Age (years)	38.0 ± 6.7 (IQR = 4.0)	39.8 ± 9.8 (IQR = 9.0)	40.2 ± 6.6 (IQR = 11.0)	36.1 ± 6.6 * (IQR = 8.0)
Height (cm)	178.3 ± 8.3 (IQR = 7.0)	179.4 ± 8.6 (IQR = 11.0)	180.4 ± 6.9 (IQR = 9.0)	180.3 ± 6.5 (IQR = 9.0)
Weight (kg)	81.7 ± 13.3 (IQR = 16.5)	87.7 ± 13.6 (IQR = 19.6)	93.8 ± 15.3 * (IQR = 15)	83.7 ± 12.5 (IQR = 16.0)

* Statistically significant difference at *p* < 0.05. IQR—interquartile range.

**Table 3 ijerph-19-06828-t003:** Average energy expenditure of men studying in the police academy during an 8-h training day.

N = 50	Lectures, Tactics Classes, Detention (Handcuffing), Physical Education Classes							
		X	±	SD	Median	Min	Max	IQR
	Time (h)	9.5	±	2.5	9.4	5.6	25.2	0.8
	kcal/h	279	±	68	261	141	441	92
Energy expenditure	kcal/min	4.65	±	1.13	4.35	2.35	7.35	1.5
	kcal/h/kg bw	3.3	±	0.9	3.2	1.3	5.4	1.1
	kcal/min/kg bw	0.055	±	0.014	0.052	0.022	0.09	0
	kcal measured	2646.8	±	1007.7	2602.5	1461	5117	1016
	Max	169.4	±	24.7	170.5	93	220	32.0
Pulse	Min	59.5	±	11.8	61	47	77	11.0
	Average	92.6	±	8.2	94	68	108	12.0
**Average**	**kcal/8 h**	**2233**	**±**	**546**	**2247**	**1127**	**3527**	**736**

IQR—interquartile range.

**Table 4 ijerph-19-06828-t004:** Energy burden related to implementation of training at specialist courses at the police training center.

N = 74	Police Training Center							
		X	±	SD	Median	Min	Max	IQR
Energy expenditure	Time	6.3	±	1.1	6.3	1.8	7.6	0.47
	kcal/h	299.9	±	74.4	299.6	105.2	460.6	96.14
	kcal/min	4.99	±	1.24	4.99	1.75	7.67	1.60
	kcal/h/kg bw	3.4	±	1	3.1	1.4	6.3	1.31
	kcal/min/kg bw	0.056	±	0.016	0.052	0.023	0.102	0.02
	kcal measured	1889.8	±	964.7	1884.5	656	3402	683
Pulse/min	Maximum	166	±	28	170	47	232	34
	Minimum	61	±	103	52	32	89	11
	Average	97.1	±	9.3	94	83	125	12
**Average**	**kcal/8 h**	**2458**	**±**	**723**	**2242**	**1010**	**4546**	**769.1**

IQR—interquartile range.

**Table 5 ijerph-19-06828-t005:** Energy load related to the training of officers—future service dog handlers.

N = 18	Training Center for Service Dog Handlers for the Police							
		X	±	SD	Median	Min	Max	IQR
Energy expenditure	Time (h)	5.3	±	1.3	5.6	0.8	6.1	0.5
	kcal/h	263.8	±	104.2	263.7	1388.4	448.7	213.87
	kcal/min	4.39	±	1.73	4.39	2.30	7.47	3.56
	kcal/h/kg bw	3.10	±	1.00	3.10	1.60	5.60	1.34
	kcal/min/kg bw	0.052	±	0.017	0.052	0.026	0.092	0.02
	kcal measured	1422	±	727	1299	212.0	2737	1441
Pulse/min	Maximum	56.4	±	24.4	152,0	126.0	228.0	37
	Minimum	60.0	±	11.0	63.0	39.0	75.0	13
	Average	94.0	±	9.8	96.0	78.0	109.0	19
**Average**	**kcal/8 h**	**2111**	**±**	**834**	**2110**	**1107**	**3590**	**857** **.98**

IQR—interquartile range.

**Table 6 ijerph-19-06828-t006:** Energy burden of officers related to the training of policemen performing tasks on water.

N = 21	Training of Policemen Performing Tasks at the Water Areas							
		X	±	SD	Median	Min	Max	IQR
Energy expenditure	Time (h)	6.0	±	0.0	6.0	6.0	6.0	0.13
	kcal/h	221.7	±	69.1	230.9	108.7	415.8	63.14
	kcal/min	3.69	±	1.73	3.84	1.81	6.93	1.05
	kcal/h/kg bw	2.5	±	0.9	2.5	1.3	5.1	1.07
	kcal/min/kg bw	0.042	±	0.014	0.041	0.021	0.085	0.02
	kcal measured	1433.6	±	441.9	1470.0	712.0	2668.0	387
Pulse/min	Maximum	144.7	±	20.8	143.0	113.0	192.0	23
	Minimum	60.0	±	10.0	58.0	43.0	81.0	11
	Average	85.4	±	10.7	87.0	65.0	107.0	9
**Average**	**kcal/8 h**	**1973**	**±**	**553**	**1847**	**870**	**3326**	**505** **.12**

IQR—interquartile range.

**Table 7 ijerph-19-06828-t007:** Energy expenditure of male officers trained at the police school.

N = 47	Training of Policemen at the Police School							
		X	±	SD	Median	Min	Max	IQR
Energy expenditure	Time (h)	7.0	±	2.70	6.5	2.5	24.0	1.5
	kcal/h	283.0	±	118.0	292.0	54.0	552.0	166.6
	kcal/min	4.71	±	1.96	4.86	0.90	9.20	2.8
	kcal/h/kg bw	3.0	±	1.2	2.8	0.7	5.7	2.0
	kcal/min/kg bw	0.050	±	0.020	0.047	0.011	0.095	0
	kcal measured	1976.0	±	947.0	1982.0	345.0	4930.0	1248
Pulse/min	Maximum	159.6	±	27.8	155.0	115.0	224.0	28
	Minimum	66.2	±	11.2	66.0	27.0	83.0	16
	Average	92.9	±	11.4	93.0	66.0	116.0	16
**Average**	**kcal/8 h**	**2267**	**±**	**942**	**2334**	**936**	**4415**	**1332** **.6**

IQR—interquartile range.

**Table 8 ijerph-19-06828-t008:** Energy expenditure of officers trained in the police prevention unit.

N = 60	Training of Policemen at Police Prevention Unit							
		X	±	SD	Median	Min	Max	IQR
Energy expenditure	Time (h)	5.7	±	1.0	6.0	2.6	7.0	0.88
	kcal/h	380.0	±	164.0	381.0	92.0	678.0	267
	kcal/min	6.33	±	2.73	6.35	1.53	11.3	4.45
	kcal/h/bw	4.6	±	1.9	4.9	1.3	8.2	3.43
	kcal/min/bw	0.076	±	0.031	0.082	0.021	0.138	0.06
	kcal measured	2171.0	±	1013.9	2170.0	244.0	4464.0	1412
Pulse/min	Maximum	172.4	±	26.1	179.0	115.0	226.0	42
	Minimum	62.9	±	11.1	64.0	44.0	78.0	15
	Average	102.7	±	16.8	106.0	62.0	134.0	26
**Average**	**kcal/8 h**	**3043**	**±**	**1308**	**3048**	**937**	**5427**	**2136**

IQR—interquartile range.

**Table 9 ijerph-19-06828-t009:** Average values of energy expenditure of the police officers trained within 8 h.

N = 280	The Average Values of Energy Expenditure Related to the Training of Police Officers							
		X	±	SD	Median	Min	Max	IQR
Energy expenditure	Time (h)	7.0	±	2.5	6.8	0.8	25.6	1.52
	kcal/h	297.0	±	118.0	287.0	54.0	678.0	146.6
	kcal/min	4.95	±	1.96	4.78	0.90	11.3	2.44
	kcal/h/bw	3.50	±	1.40	3.20	0.70	8.20	1.76
	kcal/min/kg bw	0.058	±	0.023	0.053	0.011	0.136	0.03
	kcal measured	2035.0	±	910.1	1954.0	212.0	5117.0	1118
Pulse/min	Maximum	165.4	±	25.4	165.5	93.0	232.0	38
	Minimum	62.2	±	11.2	63.0	47.0	89.0	14
	Average	95.7	±	12.6	94.5	62.0	137.0	17
**Average**	**kcal/8 h**	**2314**	**±**	**945**	**2294**	**736**	**5427**	**1173**

IQR—interquartile range.

**Table 10 ijerph-19-06828-t010:** Differences in the value of energy expenditure related to the process and specificity of training students at police schools and training centers.

Energy Expenditure	Students at the Police Academy	Students at the Police Training Center			Students at the Police School	Policemen Trained in the Police Prevention Department
		Specialist Courses	Dog Handlers	Water Police Training		
kcal/min/kg bw	0.055 ± 0.014	0.056 ± 0.016	0.052 ± 0.017	0.042 ± 0.014 *	0.050 ± 0.020	0.076 ± 0.031
Daily energy expenditure kcal/8 h	2223 ± 546	2458 ± 723	2111 ± 834	1973 ± 553 *	2267 ± 942	3043 ± 1308
Classification of the severity of work	Very hard work	Very hard work	Very hard work	Hard work	Very hard work	Very hard work

* Statistically significant difference at *p* < 0.05.

## Data Availability

The data presented in this study are available on request from the corresponding author.

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
