# Peer review of "Assessment of Energy Expenditure of Police Officers Trained in Polish Police Schools and Police Training Centers"

_ijerph, 2022, doi:10.3390/ijerph19116828_

Round 1

Reviewer 1 Report

In this paper, the authors carried out an assessment of energy expenditure of police officers trained in Polish police schools and police training centers. There is no doubt that the work carried out by the authors is of great practical value for the police officers training. However, However, from a research paper perspective, this paper has much room for improvement. The current version is more of a statistical report than an academic paper.More detailed comments are as follows:

  1. In the section of Introduction, I did not see any descriptions of the related work carried out by peers, as this part is important for the motivation of this paper.
  2. In the section of Methods, The authors have only done a simple statistical analysis (e.g., SD, Median, etc), more correlation analysis is missing.  
  3. Lack of more explanation and analysis of the data in the tables. For example, in line 107, "Lack of more explanation and analysis of the data in the tables", such a sentence does not provide the reader with any help in accessing the information in the table. 

Author Response

In the introduction, relevant publications on the energy expenditure of students of military universities and fire schools have been added.

In the methodology part, a study of statistically significant differences in relation to anthropometric indicators and the value of energy expenditure related to the training program in individual universities were added (table 2 and 10).

Explanations of the data in the tables have been added.

Reviewer 2 Report

This journal was written to a very good standard. The Abstract was well written, the Methodology covered all the important aspects, the findings was well explained and the use of tables amplified the statistics very well. The conclusion brought all the key points and answered the research question.

Minor improvement needed.

Numbers one to nine are written in words and thereafter in figures. There were some minor errors with this academic principle.

Abstract - line 25

Introduction - line 47

Materials and Methods - line 69, 70, 78, 97

Results - line 186

Numbers one to nine are written in words and thereafter in figures. There were some minor errors with this academic principle.

Author Response

The errors indicated by the reviewer have been corrected.

Reviewer 3 Report

It is so difficult to figure out how many subjects were used as described in line 70~73. The following sentence is much better to understand based on the Table 2. “The research involved 50 male and 10 female students of the Police Academy, 113 male students trained on specialist courses in the Police Training Center, 47 male students of the Police School and 60 male police officers trained in the Police Prevention Department.”

It is better to use a unified gender classification as a whole. For example, male vs female or men vs women. Please do not use all of them.

It is not clear where 74 subjects described in table 5 and 18 subjects stated in table 6 came from. This should be mentioned in the methods (participants section), maybe in line 78-79 with parenthesis.

First of all, in terms of population, too many groups are included in the study. The purpose of the study is to measure the energy expenditure based on the degree of police training and work-related, but comparing them with groups that receive a lot of physical training such as police academy and police school seems problematic. Rather, it seems that the purpose of the study is to measure the energy consumption according to the police organization rather than the police work. In addition, the number of female police officers is only 10 out of the total number of 280 subjects, so I do not know if it is meaningful to include this.

The authors need to reorganize the manuscript as follows. First, classifying into the four groups (police academy, police training center, police school and police prevention department) shown in the study, classifying them into training status and work-related energy consumption (as in Table 1) in each group, and then showing how many subjects belong to the category according to energy consumption in each group.

Author Response

In the section "Participants", the students were divided into four groups with the number of participants in each group.

The names of gender were standardized.

The aim of the work is to determine the energy expenditure of students of the police academy, police school and police training center, as well as people trained as part of the courses in the police prevention department, and to determine the degree of the heaviness of work related to the education process and to indicate the differences in the energy load of students depending on the profile and specificity training. Four institutions educating police officers are included in the materials and methods subsection, with the number of people participating in the study. Moreover, as suggested by the reviewer, the participation of women in the study was omitted.

Table 10 presents the energy expenditure values resulting from the program's implementation on a typical training day in each group of respondents and the category of work intensity.

Round 2

Reviewer 1 Report

Most of my suggestions have been responded to in the text of the paper The clarity of the paper has increased. I only have the following comment for the provided second version of the manuscript: In addition to indicators such as standard deviation, mean, etc., authors may also consider using IQR(Interquartile Range) for analysis, please see https://doi.org/10.1080/13658816.2020.1833016.  

Author Response

Dear reviewer, Thank you very much for your valuable comments. In accordance with the comments,
IQR values ​​have been added to the tables.
